# Structure of the human Bre1 complex bound to the nucleosome

Shuhei Onishi [1], Kotone Uchiyama [1], Ko Sato[1], Chikako Okada[1], Shunsuke Kobayashi[1], Keisuke Hamada[1], Tomohiro Nishizawa [2,3], Osamu Nureki [2], Kazuhiro Ogata[1] ✉ & Toru Sengoku [1] ✉

Histone H2B monoubiquitination (at Lys120 in humans) regulates transcription elongation and DNA repair. In humans, H2B monoubiquitination is catalyzed by the heterodimeric Bre1 complex composed of Bre1A/RNF20 and Bre1B/RNF40. The Bre1 proteins generally function as tumor suppressors, while in certain cancers, they facilitate cancer cell proliferation. To obtain structural insights of H2BK120 ubiquitination and its regulation, we report the cryo-electron microscopy structure of the human Bre1 complex bound to the nucleosome. The two RING domains of Bre1A and Bre1B recognize the acidic patch and the nucleosomal DNA phosphates around SHL 6.0–6.5, which are ideally located to recruit the E2 enzyme and ubiquitin for H2BK120-specific ubiquitination. Mutational experiments suggest that the two RING domains bind in two orientations and that ubiquitination occurs when Bre1A binds to the acidic patch. Our results provide insights into the H2BK120-specific ubiquitination by the Bre1 proteins and suggest that H2B monoubiquitination can be regulated by nuclesomal DNA flexibility.

Histones undergo various posttranslational modifications such as methylation, acetylation, and ubiquitination of lysine, which collectively regulate essentially all aspects of genome functions[1]. These modifications induce varied biological outputs depending on their chemical types and locations. Moreover, some histone modifications are known to affect the installation, removal, and biological outputs of other modifications, known as "crosstalk" between modifications[2].

Ubiquitination of lysine residues is a major histone modification that regulates various genomic processes such as transcription, replication, and DNA repair[3]. In humans, histones are monoubiquitinated at multiple lysine residues, including H3K14, H3K18, H3K23, H2AK13, H2AK15, H2AK119, and H2BK120 (corresponding to H2BK123 in yeast). Among them, monoubiquitination of H2BK120 (H2BK120ub) and H2AK119 (H2AK119ub) are involved in transcriptional regulation but play opposite roles; H2BK120ub positively regulates transcription[4], whereas H2AK119ub mediates transcriptional repression via the

formation of facultative heterochromatin by polycomb group proteins[5]. H2BK120ub is also important for the repair and replication of genomic DNA[4]. Mechanistically, H2BK120ub stimulates the catalytic activity of histone H3K4 and H3K79 methyltransferases (Dot1L and MLL/Set1 family complexes, respectively)[6–13], thus playing a central role in a modification crosstalk that establishes a transcriptionally active chromatin environment. Moreover, H2BK120ub itself induces an accessible, open chromatin conformation[14] and recruits other chromatin regulators such as FACT (a histone chaperone)[15] and the Swi/Snf complex (a chromatin remodeling ATPase)[16] to regulate transcription elongation.

In yeast, H2BK123 is ubiquitinated by the Bre1 protein, which forms a heterodimer and acts as an E3 ubiquitin ligase enzyme[17,18]. In humans, two homologs of yeast Bre1 (Bre1A, also known as RNF20, and Bre1B, also known as RNF40) form a heterodimer responsible for H2BK120 ubiquitination, with Rad6A as the E2 enzyme[10]. The three Bre1 proteins share a C-terminal RING domain (Fig. 1a), which is

[1]Department of Biochemistry, Yokohama City University Graduate School of Medicine, Yokohama, Japan. [2]Department of Biological Sciences, Graduate School of Science, The University of Tokyo, Tokyo, Japan. [3]Present address: Graduate School of Medical Life Science, Yokohama City University, Yokohama, Japan. ✉e-mail: ogata@yokohama-cu.ac.jp; tsengoku@yokohama-cu.ac.jp

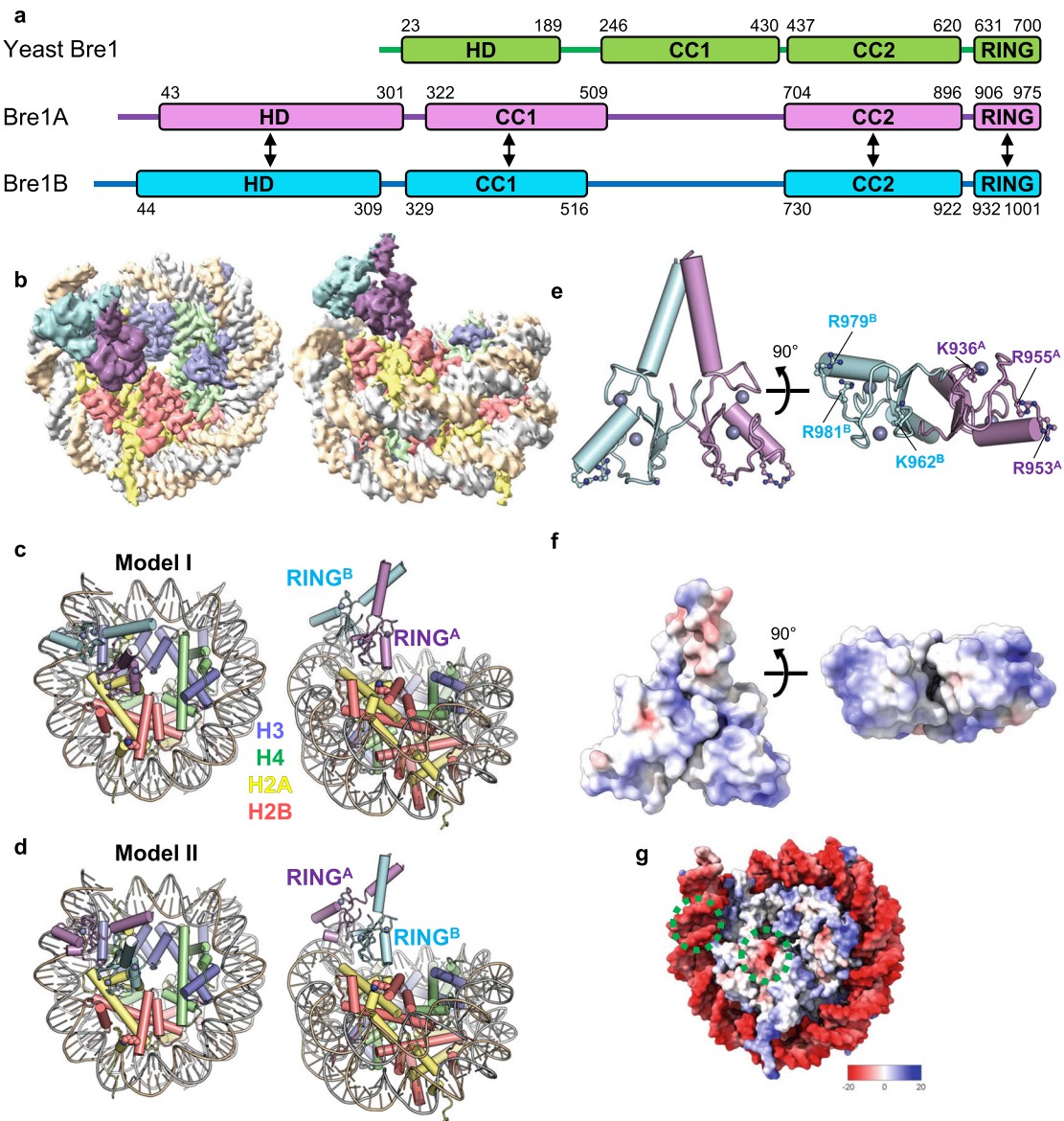

**Fig. 1 | Overall structure. a** Domain structures of yeast Bre1 and human Bre1A and Bre1B. An N-terminal helical domain (HD) and two coiled-coil regions (CC1 and CC2) are predicted based on the model structures calculated using AlphaFold. The residue numbers of the domain boundaries are shown. The arrows indicate the regions predicted to form intermolecular interactions. **b** Cryo-EM density map in two views, colored in accordance with the model in (**c**). **c**, **d** Atomic models of the two RING domains of Bre1A and Bre1B bound to the nucleosome in two views. In (**c**) (Model I) and (**d**) (Model II), RING$^A$ and RING$^B$ bind to the acidic patch, respectively. **e** Structure of the RING domain heterodimer (two views). Six basic residues that are important for nucleosome binding are shown. Zinc ions are shown as a sphere model. **f** Electrostatic surface potential of the RING domain heterodimer (two views). **g** Electrostatic surface potential of the nucleosome. The two Bre1-interaction interfaces (the acidic patch and the DNA backbone near SHL6.0) are indicated by green circles.

required for the ubiquitination of H2BK120 in humans and H2BK123 in yeasts[17–20]. In yeast Bre1, the RING domain is minimally sufficient for nucleosomal H2BK123 ubiquitination[21]. Its basic residues are important for the activity[22], presumably via interactions with the so-called "acidic patch", a cluster of acidic residues from H2A and H2B and the hotspot of nucleosome recognition by various chromatin factors[23]. Structural studies have demonstrated that two histone H2A-specific ubiquitin ligases, the RING1B-Bmi1 complex (ubiquitinating H2AK119) and the BRCA1-BARD1 complex (ubiquitinating H2AK125, K127, and K129), recognize the acidic patch through their basic residues in the RING domains of catalytic subunits[24–26]. On the other hand, although the crystal structures of the RING domain of the yeast Bre1 homodimer[27] and human Bre1A[28] have been determined, structural insights into the H2BK120-specific ubiquitination by the Bre1 proteins have been elusive.

Accumulating evidence suggests important and complicated roles of the human Bre1 complex and H2BK120ub in cancers[29]. In many types of cancers, expression of Bre1A or Bre1B is downregulated, and a decrease in global H2BK120ub level is associated with poor prognosis[30–40]. Thus, Bre1A and Bre1B are generally considered tumor suppressors. On the other hand, in luminal breast cancer cells, Bre1A silencing reduces the proliferation and tumorigenic capacity of the cells, which suggests that it plays a tumor-promoting role[31]. Bre1A expression is also required for the development of MLL-rearranged leukemia, possibly because it facilitates the transcription elongation of leukemogenic genes driven by MLL-fusion proteins[41]. Interestingly, the knockdown of Bre1A, but not Bre1B, inhibited the proliferation of MLL-rearranged leukemia cells[41]. This result indicates the paralog-specific roles of human Bre1 proteins in cancer, but it has not been investigated whether the two Bre1 proteins have asymmetric functions.

**Table 1 | Cryo-EM data acquisition, refinement, and validation statistics**

| Data collection | | | |
|---|---|---|---|
| Magnification | 81,000 | | |
| Voltage (kV) | 300 | | |
| Electron exposure (e⁻/Å²) | 58 | | |
| Frame | 54 | | |
| Defocus range (μm) | −0.8 to −1.8 | | |
| Pixel size (Å) | 1.05 | | |
| No. of micrographs | 4653 | | |
| Data processing | Bre1 complex | | Free nucleosome |
| EMDB ID | EMD-34274 | | EMD-34275 |
| Symmetry imposed | $C_1$ | | $C_1$ |
| Particles | 83,000 | | 327,589 |
| Map resolution (Å) <br> FSC threshold | 2.81 <br> 0.143 | | 2.51 <br> 0.143 |
| Refinement | Model I | Model II | Free nucleosome |
| PDB ID | 8GUI | 8GUJ | 8GUK |
| Model composition | | | |
| Non-hydrogen atoms | 13348 | 13348 | 12098 |
| Protein residues | 920 | 920 | 765 |
| DNA residues | 294 | 294 | 294 |
| Zinc ions | 4 | 4 | 0 |
| Model vs. data CC (mask) | 0.85 | 0.85 | 0.86 |
| Mean $B$ factors (Å²) | | | |
| Protein | 63.11 | 63.63 | 70.06 |
| DNA | 73.95 | 72.74 | 56.87 |
| RMS deviations | | | |
| Bond lengths (Å) | 0.004 | 0.004 | 0.004 |
| Bond angles (°) | 0.631 | 0.630 | 0.634 |
| Validation | | | |
| MolProbity score | 1.49 | 1.48 | 1.63 |
| Clash score | 6.55 | 7.41 | 5.13 |
| Ramachandran plot | | | |
| Favored (%) | 97.33 | 97.67 | 98.13 |
| Allowed (%) | 2.67 | 2.33 | 1.87 |
| Outliers (%) | 0.00 | 0.00 | 0.00 |

To obtain structural insights into H2BK120-specific ubiquitination by Bre1 complexes and gain insights into its regulation and paralog-specific roles, we report the cryo-electron microscopy (cryo-EM) structure of the human Bre1 complex bound to the nucleosome. The structure reveals that the RING domains of Bre1A and Bre1B directly interact with the acidic patch and the DNA phosphates of the nucleosome through their basic residues. One RING domain exhibits flexible interaction with the nucleosomal DNA, suggesting that H2BK120 ubiquitination may be regulated by histone modifications that affect DNA flexibility. Mutational analyses reveal that the Bre1A subunit acts as the catalytic E3 enzyme by recruiting Rad6A and identify key residues for Rad6A binding. The orientation and location of the two RING domains are consistent with the H2BK120 specificity of the Bre1 complex but differ markedly from those of the H2A-specific ubiquitin ligases. Our study provides a structural framework for the regulation of H2BK120 ubiquitination by the Bre1 complex.

## Results
### Overall structure
We reconstituted the trimeric Bre1A-Bre1B-Rad6A by mixing the coexpressed Bre1A-Bre1B heterodimer and individually expressed Rad6A and confirmed its nucleosome-binding and nucleosomal H2BK120 ubiquitination activities (Supplementary Fig. 1). Using this trimeric complex and in vitro assembled nucleosome with 147-bp DNA, we determined the cryo-EM structure of the Bre1 complex bound to the nucleosome (Supplementary Figs. 2–4, Table 1, and Supplementary Movie 1). When we calculated a reconstruction map from cryo-EM images at 2.8 Å using a conventional method, the density corresponding to the two Bre1 proteins were blurred (local resolution > 5 Å), suggesting a flexible binding mode of the Bre1 complex to the nucleosome. To better resolve the density of the Bre1 complex, we applied 3D Flexible Refinement (3DFlex) analysis[42], which uses a motion-based deep neural network model of continuous heterogeneity. We obtained a map (the 3DFlex map) with an improved local resolution of the Bre1 complex (around 4.0 Å) with the latent space dimensionality $K = 2$, while the overall resolution did not improve. We also determined the structure of the free nucleosome at 2.51 Å resolution as a reference (Fig. 1, Supplementary Figs. 2, 3, and Table 1). Although we used the full-length Bre1A-Bre1B-Rad6A heterotrimer for grid preparation, only the C-terminal RING domains of Bre1A and Bre1B (hereafter designated as RING^A and RING^B, respectively) were visible in the reconstructed cryo-EM maps (Fig. 1a, b). This agrees with the model structures of the yeast and human Bre1 complexes calculated by AlphaFold[43–45], in which the C-terminal RING domains are preceded by short flexible linkers and appear to move flexibly with respect to the other regions of the complex (Supplementary Fig. 5). Previous studies have reported that the yeast and human Bre1 complexes bind Rad6 proteins via their N-terminal regions[13,21], which explains why the density of Rad6A was not observed in the current EM maps.

Consistent with the previous biochemical analysis of the yeast Bre1 complex[22], one RING domain of the human Bre1 complex binds to the acidic patch (Fig. 1b–d). Unexpectedly, the other RING domain binds to the DNA phosphates of one DNA strand at SHL 6.0–6.5 (Fig. 1b–d). RING^A and RING^B have positively charged molecular surfaces (Fig. 1e, f); thus, they bind to the negatively charged acidic patch and DNA phosphate groups (Fig. 1g) through electrostatic interactions. Bre1 binding does not induce a large conformational change in the nucleosome, but a region of the nucleosomal DNA that contacts Bre1 undergoes a small shift (approximately 3 Å) (Supplementary Fig. 6). Since the local resolution of this DNA region is relatively low, as often observed in nucleosome structures, the functional significance of this small shift is unclear.

As the sequences of RING^A and RING^B are highly similar (with 86% identity; Supplementary Fig. 7), we were unable to assign the orientation of the two subunits to the density. Thus, we created two atomic models of the RING^A-RING^B complex bound to the nucleosome based on the obtained density. In Model I (Fig. 1c), RING^A and RING^B bind to the acidic patch and DNA phosphates, respectively, whereas in Model II (Fig. 1d), the two RING domains have opposite orientations. Based on our biochemical experiments, we propose that the Bre1 complex can bind to the nucleosome in both orientations and that Model I, in which RING^A binds to the acidic patch, represents the catalytically active form for H2BK120 ubiquitination (see below).

### Bre1 activity may be regulated by the flexibility of the nucleosomal DNA
The 3DFlex analysis not only provides a density map with higher local resolution, but it also reveals flexible motions. Briefly, the flexible three-dimensional structure of a protein is represented as deformations of a single canonical density map, using a latent coordinate and a neural flow generator. This deformation results in a convected map that corresponds to a single conformation of the protein. The latent dimensions generated by the 3DFlex analysis represent possible structural flexibility, or motions, of the protein.

Figure 2a shows the distribution of the latent coordinates of the particles used for the 3DFlex analysis, and Fig. 2b–e show the atomic models constructed onto the convected densities on each side of the

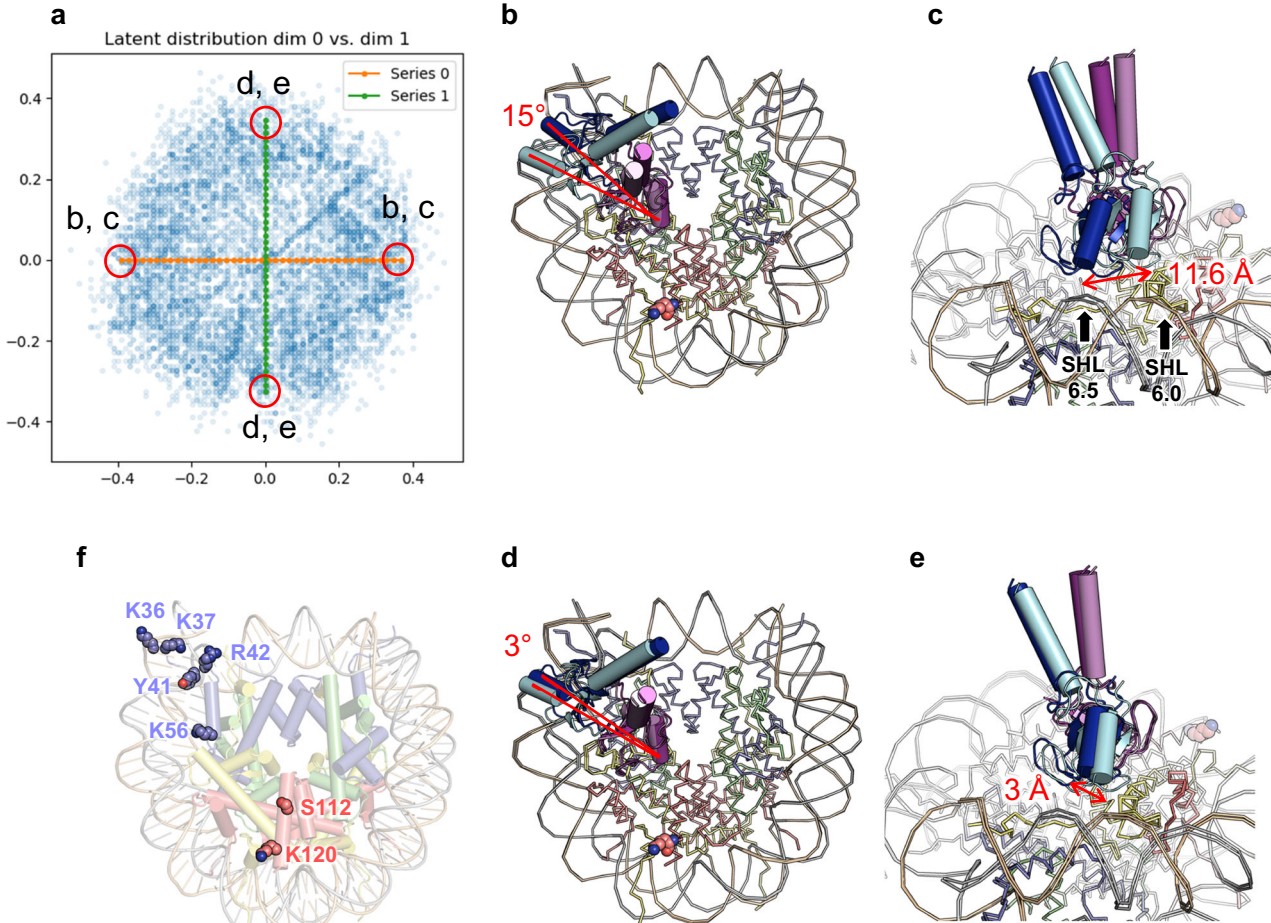

**Fig. 2 | Flexible binding of the Bre1 complex revealed by the 3DFlex analysis.**
**a** Distribution of the latent coordinates. Coordinates that correspond to the structures shown in (**b**–**e**) are indicated. Green and orange circles represent the volume series shown in Supplementary Movies 2 and 3, respectively.

**b**–**e** Conformational changes represented by the first (**b**, **c**) and second (**d**, **e**) dimensions. **b**, **d** An overall view. **c**, **e** A close-up view. **f** Residues whose modifications or substitutions (H2BS112GlcNAc, H3K36M, H3K37ac, H3Y41ph, H3R42A, H3K56ac, and H3R42me2a) upregulate Bre1 activity. H2BK120 itself is also shown.

latent coordinate axes. Supplementary Fig. 8 shows the corresponding densities, and Supplementary Movies 2 and 3 show series of convected densities along the two axes. The two types of movements represented by the two dimensions both represent rotational movements with the region around the arginine anchor as the pivot point. The first dimension represents a larger movement, with 15° rotation of the two RING domains and about 11.6 Å displacement at the outer side of the DNA-binding subunit, while the second dimension represents a smaller movement with about 3° rotation and about 3 Å displacement (Fig. 2d, e). In contrast, the structural change of the nucleosomal DNA is minimal. This observation suggests that on binding to the nucleosome, the acidic-patch binding subunit stably associates with it, while the other subunit interacts with the nucleosomal DNA in a flexible manner.

A previous study using semi-synthesized nucleosomes identified multiple histone H3 residues that, when modified or mutated, stimulates H2BK120 ubiquitination[46]. These modifications and mutations, such as H3Y41 phosphorylation (H3Y41ph) and H3K56 acetylation (H3K56ac), are clustered near the DNA entry/exit site and tend to weaken histone-DNA contacts around the Bre1-bound site (Fig. 2f). Furthermore, addition of a linker histone, which stabilizes the conformation of the nucleosomal DNA, was shown to inhibit H2BK120 ubiquitination[46]. Based on these results, it was proposed that the local structure of the DNA entry/exit site regulates the ubiquitination activity[46]. Consistent with this, our 3DFlex analysis revealed the flexible binding of the distal Bre1 subunit to nucleosomal DNA. Taken together,

these results suggest that the modifications which weaken histone-DNA contacts have an impact on the flexibility of DNA around SHL 6.0–6.5 region, potentially facilitating the productive binding of the Bre1 complex to the DNA. However, further studies are necessary to test this hypothesis and examine whether such a regulatory mechanism operates in a physiological context.

## Binding of the acidic patch and DNA phosphates

The detailed interactions between the acidic patch and RING[A] or RING[B] are shown in Fig. 3, Supplementary Fig. 9, and Supplementary Movie 4, and the density around this region is shown in Supplementary Fig. 4. Basic residues that are homologous between RING[A] and RING[B] (Figs. 1e and 3b and Supplementary Figs. 7 and 9) mediate the interactions with the acidic patch as follows. R953[A] and R978[B] (Bre1A and Bre1B residues are indicated with superscripted "A" and "B," respectively) in Models I and II, respectively, form salt bridges with H2AE61, H2AD90, and H2AE92. These arginine residues serve as the canonical "arginine anchor" in a manner similar to that of many other chromatin factors that bind the nucleosome[23]. R955[A] in Model I and R981[B] in Model II also form salt bridges with H2AE61 and H2AE64 at the position of the so-called variant arginine type 1[23]. In addition, the side chains of K936[A] in Model I and K962[B] in Model II are located at the interface with H2B, possibly contributing to nucleosome binding by forming intermolecular interactions.

The other RING domain of the Bre1 complex binds the DNA phosphates around SHL 6.0–6.5 (Fig. 3a, c and Supplementary Fig. 9).

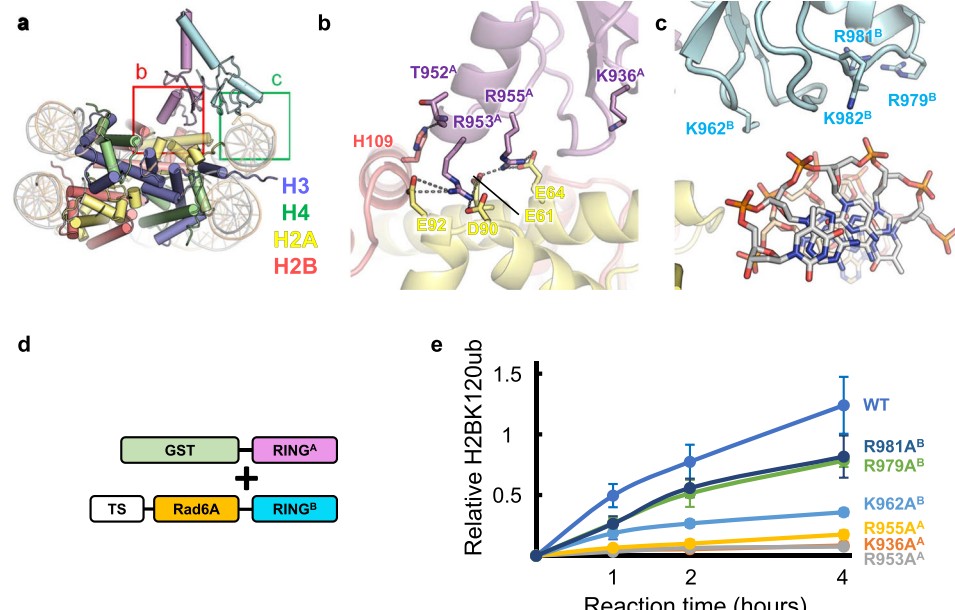

**Fig. 3 | Interactions between the Bre1 complex and the nucleosome. a** Structure of Model I. The magnified regions in (**b**, **c**) are indicated. **b** Interactions between RING[A] and the acidic patch in Model I. Hydrophilic interactions (salt bridges and hydrogen bonds) are indicated by gray dotted lines. **c** Basic residues of RING[B] near the DNA phosphates in Model I. **d** A schematic representation of the protein constructs used in the experiment shown in (**e**). TS, Twin-Strep-tag. **e** H2BK120 ubiquitination assay of the wild type (WT) and mutants possessing substitutions at basic residues. The signals were normalized usig the normalization control. The mean and standard deviation of six (for WT) or three (for mutants) independent results are shown. Source data are provided as a Source Data file.

K962[B], R979[B], R981[B], and R982[B] in Model I and K936[A], R953[A], R955[A], and K956[A] in Model II face DNA phosphates. Although the lower local resolution around this region hinders the precise modeling of their side chain atoms (Supplementary Fig. 4), these residues possibly mediate the interaction with DNA by forming salt bridges. We note that among these residues, R953[A], R955[A], R979[B], and R979[B] mediate interactions with the acidic patch when the Bre1 complex binds to the nucleosome in the opposite direction.

## Basic residues of the RING domains important for H2BK120 monoubiquitination

To examine the roles of the basic residues in RING[A] and RING[B], we prepared mutant Bre1 complexes carrying alanine mutations at the positions of K936[A], R953[A], R955[A], K962[B], R979[B], and R981[B]. To focus on the functional role of the RING domains and facilitate enzymatic analysis, we created constructs expressing RING[A] or RING[B] fused to Twin-Strep-tag and Rad6A and coexpressed each of them with the GST-fused RING domain of the other Bre1 protein (Fig. 3d and Supplementary Fig. 10a). A similar fusion strategy was used to analyze the activity of the yeast Bre1 complex[21,22]. To purify the heterodimeric complexes, the coexpressed proteins were purified by serial affinity chromatography steps using Strep-Tactin and glutathione resins and then used for enzymatic analysis.

The results of the enzymatic analysis using GST-RING[A] and Twin-Strep-Rad6A-RING[B] are shown in Fig. 3e, whereas those of the enzymatic analysis using GST-RING[B] and Twin-Strep-Rad6A-RING[A] are shown in Supplementary Fig. 10b. All six mutations showed reduced activity, albeit at varying degrees. The K936[A], R953[A], and R955[A] mutations almost completely abolished the activity in both combinations. The K962[B] and R981[B] mutations showed milder effects but still showed reduced activity in both combinations. The R979[B] mutation resulted in a mild reduction when GST-RING[A] and Twin-Strep-Rad6A-RING[B] were used, while it almost completely abolished the activity when GST-RING[B] and Twin-Strep-Rad6A-RING[A] were used. These results demonstrate the importance of the interaction between the basic residues of the two RING domains and the acidic patch and

nucleosomal DNA observed in the current structure for H2BK120 ubiquitination.

## Bre1A as a catalytic subunit

The mutational analysis results showed that mutations on Bre1B had milder effects, which suggests that Bre1A and Bre1B play asymmetrical functions in catalysis, even though their sequences are similar. Consistent with this, a previous paper noted that the ectopic expression and knockdown of Bre1A, not Bre1B, affected the H2BK120ub level in HEK293T cells[19], although the data on Bre1B was not shown in this paper. Thus, we addressed whether the RING domains of Bre1A and Bre1B may perform asymmetrical functions. Many RING-type E3 enzymes, including the H2A-specific enzymes RING1B and BRCA1, are known to bind non-catalytic RING proteins to form RING domain heterodimers, in which the catalytic RING subunit recruits the E2 enzyme and ubiquitin for site-specific ubiquitination[47]. To examine if Bre1 proteins can asymmetrically interact with its cognate E2 protein, Rad6A, we first created model structures of Rad6A bound to the Bre1 RING domains by superposing with the structures of a known RING E3-E2-ubiquitin complex, the RNF4-UbcH5A ubiquitin complex[48] (PDB ID 4AP4), followed by the superposition of human Rad6A[49] (PDB ID 6CYO) on UbcH5A (Fig. 4a, Supplementary Movie 5). Previous reports have created model structures of yeast and human Bre1 complexes with bound Rad6 in a similar manner, which were supported by cross-linking and mass spectrometry analyses or a mutational assay[22,28]. Our human Bre1-Rad6A model suggests that if RING[A] binds Rad6, the interaction may involve Bre1A residues C924[A], N926[A], M927[A], T948[A], R949[A], T952[A], Q954[A], K956[A], P958[A], and K959[A]. Bre1A and Bre1B have different amino acids only at three (M927[A]/T953[B], T948[A]/G974[B], and T952[A]/A978[B]) of these positions (Fig. 4b, Supplementary Fig. 7). Furthermore, in the model structure, the side chains of T948[A] and T952[A] are located within the hydrogen-bonding distance of the N65 and K66 side chains of Rad6A (Fig. 4c). These observations suggest that the two positions, T948[A]/G974[B] and T952[A]/A978[B], are likely located at the interaction interface with Rad6A and that the amino acids at these positions may be involved in the asymmetrical functions of Bre1A and Bre1B.

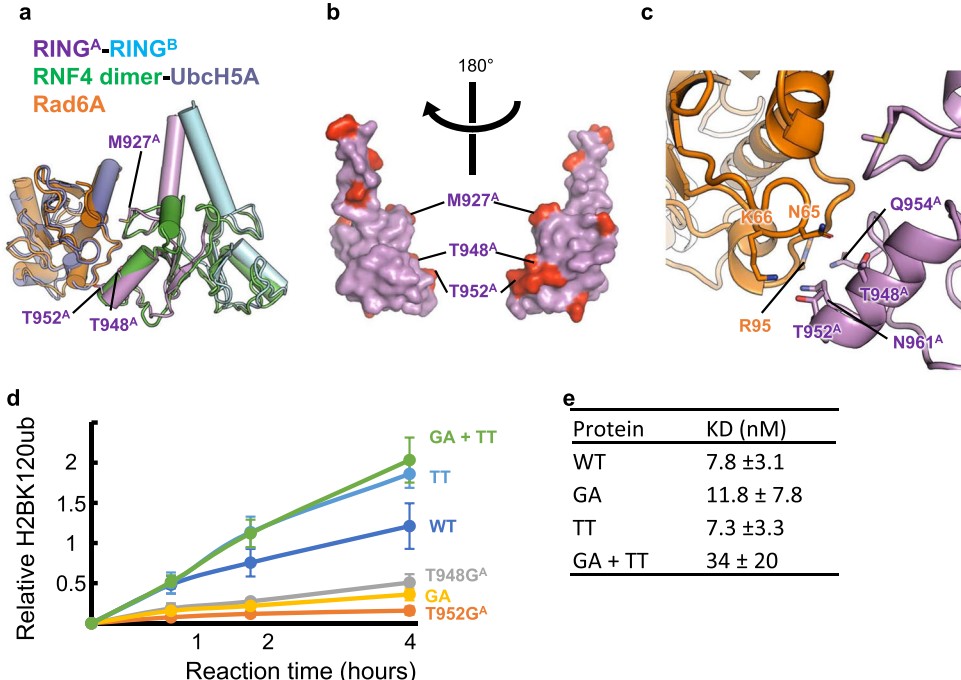

**Fig. 4 | Identification of Bre1A residues likely to mediate Rad6A binding.**
**a** Structural modeling of the RING$^A$-RING$^B$-Rad6A complex. The current structure of the RING$^A$-RING$^B$ complex is superposed on that of the RNF4 dimer bound to UbcH5A and ubiquitin (PDB ID 4AP4, ubiquitin not shown), and then the structure of Rad6A (PDB ID 6CYO) is further superposed on that of UbcH5A. **b** RING$^A$ residues that are not conserved in RING$^B$. The RING$^A$ structure is shown as a surface representation, and 10 non-conserved residues are colored red. Three residues near a possible Rad6A binding region are labeled. **c** Close-up view of the RING$^A$-RING$^B$-Rad6A model in (**b**). **d** H2BK120 ubiquitination assay of the wild type (WT) and mutants possessing substitutions at the possible Rad6A binding region. The constructs shown in Fig. 2d and their mutants were used. GA: T948G$^A$-T952A$^A$. TT: G974T$^B$-A978T$^B$. The mean and standard deviation of six (for WT) or three (for mutants) independent results are shown. Source data are provided as a Source Data file. **e** The binding affinities of the WT and mutants for the nucleosome. The constants between the Bre1 complexes and the nucleosome are shown with their 1-sigma confidence interval. The values were calculated with four (for GA + TT) or three (for the others) independent results.

To directly test whether T948$^A$/G974$^B$ and T952$^A$/A978$^B$ play important roles, we purified Bre1-Rad6 fusion complexes with the Bre1A subunit carrying a single mutation or double Bre1B-like mutations as follows: T948G$^A$, T952A$^A$, or T948G$^A$-T952A$^A$ (hereafter referred to as GA). We then analyzed their nucleosomal H2BK120 ubiquitination activity. As shown in Fig. 4d, the T948G$^A$, T952A$^A$, and GA mutants all exhibited significantly reduced activity. To further validate the importance of threonine at these positions, we created two Bre1 complexes, one containing the Bre1B subunit carrying Bre1A-like dual mutations (G974T$^B$-A978T$^B$, hereafter referred to as TT) and the wild-type Bre1A subunit, and the other carrying Bre1B-like Bre1A and Bre1A-like Bre1 subunits (GA + TT). As shown in Fig. 4d, the both mutant complexes exhibited slightly higher activity than the wild-type complex. We speculate that the effect of the inactivating GA mutation in the GA + TA mutant was masked by the activating TT mutation, which for some unknown reason endowed Bre1B with a higher activity than the wild-type Bre1A. We further measured the nucleosome affinities of these mutations by microscale thermophoresis and found that the wild-type complex and the GA and TT mutants had comparable affinities, while the affinity of the GA + TT mutant was reduced (Fig. 4e and Supplementary Fig. 11). These results show that the higher activities of the TT and GA + TT mutants are not the result of the higher nucleosome affinities. Collectively, our mutational analyses strongly suggest that Bre1A functions as a catalytic subunit to recruit the E2-ubiquion complex for H2BK120 mono ubiquitination and that both T948$^A$ and T952$^A$ are important for Rad6A binding.

## A model for H2BK120-specific ubiquitination
Figure 5a and Supplementary Movie 5 show the model structure of the RING$^A$-RING$^B$-Rad6A ubiquitin bound to the nucleosome based on

Model I, where RING$^A$ binds the acidic patch. In this model, the lysine residue closest to ubiquitin G76 is H2BK120 (Fig. 5b, Supplementary Movie 5), and the distance between the Cα atom of H2BK120 and the carbonyl carbon of ubiquitin G76 is 8.6 Å. The second closest lysine is H2BK116, whose Cα atom is located 10.8 Å away from the carbonyl carbon of ubiquitin G76, and might be too farther away or not ideally oriented for ubiquitin transfer (Fig. 5b). Unlike the C-terminal ubiquitination sites of H2A, which are structurally flexible, H2BK117 and H2BK120 are located on the C-terminal α-helix of H2B. The helical structure should exhibit limited backbone flexibility, which may hinder the access of H2BK117 to the ubiquitination catalytic center. Taken together, our model structure of the RING$^A$-RING$^B$-Rad6A-ubiquitin complex bound to the nucleosome is consistent with the H2BK120 specificity of the human Bre1 complex.

GlcNAcylation at H2BS112 was shown to facilitate H2BK120 ubiquitination by the Bre1 complex[46,50]. In our model H2BS112 is located close to Rad6A (Fig. 5b). Thus, H2BS112GlcNAc may facilitate H2BK120 ubiquitination by interacting with Rad6, thereby increasing the binding affinity or correctly positioning Rad6A and ubiquitin suitable for ubiquitination.

Our structural model also suggests that the catalytic RING domain should bind to the acidic patch to position the E2 enzyme and ubiquitin near H2BK120 for ubiquitination, as the other RING domain contacting the DNA phosphates is farther away from H2BK120 and faces the opposite direction. Thus, when the wild-type Bre1 complex ubiquitinates H2BK120, the catalytically active RING$^A$ should bind to the acidic patch. Meanwhile, the results of our mutational experiments also showed that when Bre1A-type mutations (G974T$^B$ and A978T$^B$) were introduced, RING$^B$ could function as the catalytic E3 enzyme. This suggests that RING$^B$ also binds to the acidic patch, which agrees with

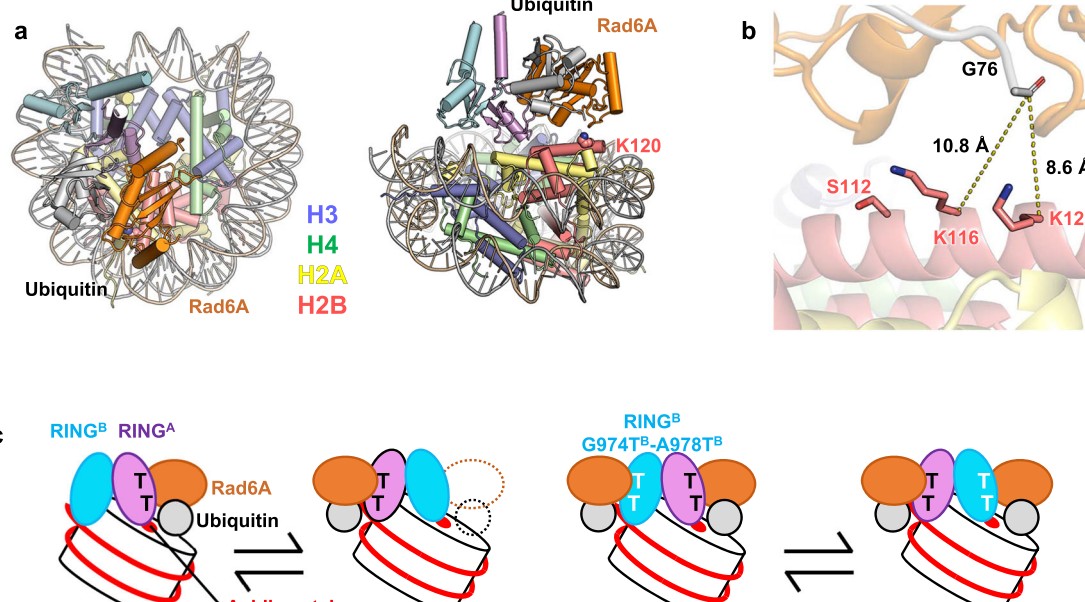

**Fig. 5 | A model for H2BK120-specific ubiquitination by the Bre1 complex.**
**a** Model structure of the RINGA-RINGB-Rad6A-ubiquitin complex bound to the nucleosome in two views. **b** Close-up view of ubiquitin and H2B. Two lysine residues (H2BK120 and H2BK116) near G76 of ubiquitin are shown. H2BS112, whose GlcNAcylation stimulates H2BK120 ubiquitination, is also shown. **c** Proposed mechanistic model. The wild-type Bre1 complex can bind to the nucleosome in two orientations, but H2BK120 ubiquitination occurs only when Bre1A binds to the acidic patch, as RING^A, but not RING^B, can recruit Rad6A and ubiquitin. Bre1B with G974T^B-A978T^B double substitution can recruit Rad6A and ubiquitin; thus, H2BK120 ubiquitination occurs in both binding modes.

the high sequence similarity between RING^A and RING^B and the perfect conservation of the two arginine residues that binds to the acidic patch (R953^A/R979^B and R955^A/R981^B).

Based on these observations, we propose a mechanistic model of how the human Bre1 complex ubiquitinates H2BK120 (Fig. 5c). The Bre1 complex interacts with the nucleosome in at least two regions: the acidic patch and the DNA phosphates around SHL 6.0–6.5. The two regions are recognized via the basic surfaces of the two RING domains. The Bre1 complex interacts with the nucleosome in two possible orientations of their RING domains, and H2BK120 ubiquitination occurs only when RING^A binds to the acidic patch. R953^A and R955^A are the canonical arginine anchor and variant arginine type 1, respectively, for recognizing the acidic patch and are thus essential for H2BK120 ubiquitination. On the other hand, basic residues in RING^B, such as K962^B, R979^B, and R981^B, have a rather supportive role in orienting the catalytic RING^A domain and providing additional affinity for the nucleosome via interaction with the nucleosomal DNA phosphates. When Bre1A-type mutations (G974T^B and/or A978T^B) are introduced, the mutant Bre1 complex can catalyze H2BK120 ubiquitination in both orientations, as both RING domains can recruit Rad6A and ubiquitin.

## Discussion

The structural comparison between the current Bre1 complex (Supplementary Fig. 12a) and two other histone ubiquitin ligases recognizing the nucleosome (the RING1B-Bmi1 complex ubiquitinating H2AK119, shown in Supplementary Fig. 12b, and the BRCA1-BARD1 complex ubiquitinating H2AK125, K127, and K129, shown in Supplementary Fig. 12c) revealed a clear similarity and difference between their nucleosome recognition mechanisms. In all structures, the catalytic RING domain recognizes the acidic patch using the conserved arginine residues as the arginine anchor, clarifying a common mechanism adopted by all structurally characterized histone ubiquitin ligases that recognize the nucleosome. In contrast, the orientation of the catalytic RING domain in the Bre1 complex differs substantially from that of the two H2A ubiquitin ligases, consistent with their

different residue specificity. Moreover, the non-catalytic RING domains interact with the nucleosome differently. Unlike the Bre1 complex, whose non-catalytic RING domain (RING^B) binds to DNA phosphates, the non-catalytic RING domains of the two H2A ubiquitin ligases (Bmi1 and BARD1) do not possess the basic surface and thus do not bind to DNA. Instead, Bmi1 contacts residues 76–79 of histone H3, whereas BARD1 interacts mainly with the H2B/H4 cleft, resulting in distinct residue specificities on the H2A C-terminal region. In summary, the two H2A ubiquitin ligases bind to nucleosomes in similar but distinct ways to specifically ubiquitinate different H2A lysine residues, whereas the Bre1 complex binds in a nearly opposite direction for H2BK120-specific ubiquitination.

To date, our understanding of the functional differences between two Bre1 proteins in human has been limited. However, a previous study on MLL-rearranged leukemia suggested they have asymmetric functions[41]. Consistently, by mutational experiment here we have identified Bre1A, but not Bre1B, as the catalytic subunit responsible for recruiting the cognate E2 enzyme (Rad6A) and ubiquitin. Supporting our findings, data from the DepMap database[51] reveals that the knockout of Bre1A has a larger impact on the survival of cancer cell lines compared to the knockout of Bre1B (average gene effect scores of −0.82 and −0.44, respectively) (Supplementary Data 1). Further studies are required to fully comprehend the distinct activities and physiological roles of the two Bre1 proteins in human under both normal and pathogenic conditions.

While this manuscript was under review, a structural study on yeast and human Bre1 complexes bound to the nucleosome was reported by Deng et al.[52]. Another study on the yeast Bre1-nucleosome complex was also reported as a preprint by Zhao et al.[53]. Deng et al. used a complex chemical trapping strategy to covalently attach the Bre1 complex and Rad6 to the nucleosome, while Zhao et al. used a fusion protein containing two tandem RING domains of Bre1 followed by Rad6. Nevertheless, the structures reported by these studies are very similar to ours using the full-length human Bre1 complex without any protein engineering; one RING subunit binds stably to the acidic

patch via arginine anchors, and the other binds flexibly to the nucleosomal DNA. The three studies complement with each other to show that the effects of a chemical trapping strategy and fusion formation on the structures are minimal, and together reveal the mechanism of specific H2B ubiquitination conserved between human and yeast. It should also be noted that Deng et al. modeled the heterodimeric Bre1 complex in one orientation with the Bre1A subunit bound to the acidic patch. Since our structural and biochemical analyses have now identified Bre1A as the likely catalytic subunit, the structure reported by Deng et al. is assumed to represent a catalytically active state, providing an insight into the Bre1A-Rad6A interactions. In this structure, T948[A] and T952[A] are located proximal to Rad6A, consistent with our proposal that these two threonine residues are responsible for Rad6A binding.

## Methods

### AlphaFold calculation

The model structures of the full-length yeast Bre1 homodimer and human Bre1 heterodimer were calculated using the program LocalColabFold[45], with AlphaFold2[43] as the calculation engine.

### Protein and DNA sequences

The protein and nucleosomal DNA sequences used in this study are shown in Supplementary Data 2.

### Purification of the Bre1A-Bre1B-Rad6A complex

The full-length human Rad6A sequence was cloned into a modified pET32 plasmid (Roche) and expressed as a fusion protein with the N-terminal thioredoxin-His6-SUMOstar tag. The plasmid was introduced in Rosetta2(DE3) pLysS cells, and the protein expression was induced by adding 0.2 mM isopropyl-β-D-thiogalactoside (IPTG) and culturing overnight at 20 °C. The harvested cells were suspended in 20 mM Na phosphate at pH 7.3, 500 mM NaCl, 1 mM DTT, 0.1 mM phenylmethylsulfonyl fluoride (PMSF), and 1% Triton X-100 and disrupted by sonication. Clarified lysate was loaded onto a cOmplete His-Tag column (Roche). The protein was eluted with 20 mM Na phosphate at pH 7.3, 500 mM NaCl, 1 mM DTT, and 500 mM imidazole. For the reconstitution of the Bre1A-Bre1B-Rad6A complex, the tag-fused Rad6A was flash-frozen in liquid nitrogen and stored at −80 °C. For the histone H2B ubiquitination assay, the tag-fused Rad6A was mixed with SUMOstar protease and dialyzed against 20 mM Tris-HCl at pH 7.5, 150 mM NaCl, and 1 mM DTT overnight. The digested sample was then mixed with an equal amount of water and loaded onto tandemly connected HisTrap HP and HiTrap Q columns (Cytiva) equilibrated with 20 mM Tris-HCl at pH 7.5, 100 mM NaCl, 20 mM imidazole, and 1 mM DTT. After disconnecting the HisTrap column, the bound Rad6A protein was eluted from the HiTrapQ column with a linear gradient from 100 mM to 1 M of NaCl in 20 mM Tris-HCl at pH 7.5, 20 mM imidazole, and 1 mM DTT. The fractions containing Rad6A were pooled, concentrated, buffer exchanged to 10 mM HEPES-Na at pH 7.5, 150 mM NaCl, 1 mM DTT, flash-frozen with liquid nitrogen, and stored at −80 °C.

For the structural study, the full-length sequences of human Bre1A and Bre1B were first individually cloned into pACEBac1 (Geneva Biotech) with the N-terminal His6-SUMOstar tag and the N-terminal Twin-Strep-His6-SUMOstar tag, respectively. The two expression cassettes were then integrated into a single pACEBac1 plasmid to construct a bicistronic vector expressing both proteins. Baculovirus production in Sf9 cells was conducted using the Bac-to-Bac system (ThermoFisher), in accordance with the manufacturer's instructions. The full-length Bre1A-Bre1B complex was expressed using the bicistronic baculovirus in High Five cells, which were collected 48 h after viral infection. The cell pellets were suspended in 20 mM Tris-HCl at pH 8.0, 200 mM NaCl, 1 mM DTT, 0.1 m PMSF, 0.5% Triton X-100, and cOmplete ULTRA protease inhibitor cocktail (Roche) and disrupted by sonication. Clarified lysate was loaded onto a Strep-Trap HP column (Cytiva), and the

complex was eluted in a buffer containing 20 mM Tris-HCl at pH 8.0, 200 mM NaCl, 1 mM DTT, and 2.5 mM desthiobiotin. To reconstitute the trimeric Bre1A-Bre1B-Rad6A complex, the eluted fractions containing the tagged Bre1A-Bre1B complex were mixed with the tag-fused Rad6A and SUMOstar protease and incubated at 4 °C overnight. The tag-removed, reconstituted ternary complex was then loaded onto a HiTrap Heparin column (Cytiva) equilibrated with 20 mM Tris-HCl at pH 7.5, 100 mM NaCl, and 1 mM DTT, and eluted with a linear gradient from 100 mM to 1 M NaCl in 20 mM Tris-HCl at pH 7.5 and 1 mM DTT. The complex was further purified with a Superose 6 column (Cytiva) equilibrated with 20 mM HEPES-Na at pH 7.5, 200 mM NaCl, and 1 mM DTT, concentrated, flash-frozen in liquid nitrogen, and stored at −80 °C.

The DNA sequences encoding the RING domain of Bre1A or Bre1B were cloned into pGEX-6P-1 (Cytiva) to express GST-RING[A] or GST-RING[B] and cloned into a modified pET plasmid (Merck) to express Twin-Strep-Rad6A-RING[A] or Twin-Strep-Rad6A-RING[B]. To prepare the RING domain heterodimers for ubiquitination assay, two plasmids (pGEX-6P-RING[A] and pET-Twin-Strep-Rad6A-RING[B] or pGEX-6P-RING[B] and pET-Twin-Strep-Rad6A-RING[A]) were simultaneously introduced into Rosetta2(DE3)pLysS cells. Protein expression was induced with 0.2 mM IPTG and overnight culture at 16 °C. The harvested cells were suspended in 40 mM K₂HPO₄, 10 mM KH₂PO₄, 500 mM NaCl, 1 mM DTT, 0.1 mM PMSF, and 0.5% Triton X-100 and lysed by sonication. The clarified lysate was applied to a StrepTrap column (Cytiva), and the bound heterodimers were eluted with 50 mM Tris-HCl at pH 7.5, 500 mM NaCl, 1 mM DTT, 2.5 mM desthiobiotin. The eluents were further purified with a GSTrap column (cytiva) and eluted with 50 mM Tris-HCl at pH 8.0, 500 mM NaCl, and 10 mM reduced glutathione. The purified heterodimers were flash-frozen in liquid nitrogen and stored at −80 °C. To express the His6-Twin-Strep-Rad6A-RING[B] for microscale themophoresis experiments, a DNA sequence encoding six histidine residues was inserted into pET-Twin-Strep-Rad6A-RING[B] The RING domain heterodimers with the His6 tag were expressed and purified in the same way as those without the His6 tag.

The plasmids encoding GST- and His-tagged human E1 (pGEX-6P-HsE1-His8) and ubiquitin (pET26b-Ub) were kind gifts from Dr. Yusuke Sato (Tottori University). pGEX-6P-HsE1-His8 was introduced into Rosetta2(DE3)pLysS cells, and protein expression was induced by adding 0.3 mM IPTG and culturing overnight at 20 °C. The harvested cells were suspended in 40 mM K₂HPO₄, 10 mM KH₂PO₄, 20 mM imidazole, 500 mM NaCl, 1 mM DTT, 0.1 mM PMSF, and 0.5% Triton X-100 and lysed by sonication. The clarified lysate was applied to a HisTrap column, and the protein was eluted with a linear gradient from 20 mM to 500 mM of imidazole in 40 mM K₂HPO₄, 10 mM KH₂PO₄, 500 mM NaCl, and 1 mM DTT. Fractions containing E1 were dialyzed against 20 mM Tris-HCl at pH 7.5 and 50 mM NaCl overnight. The dialyzed sample was applied to a HiTrap Q column (Cytiva) and eluted with a linear gradient from 50 mM to 500 mM NaCl in 20 mM Tris-HCl at pH 7.5. The eluted sample was applied to a GSTrap column; eluted with 50 mM Tris-HCl at pH 8.0, 50 mM NaCl, and 10 mM glutathione; flash-frozen in liquid nitrogen; and stored at −80 °C.

### Nucleosome reconstitution

The nucleosome was reconstituted with human core histones and 147- or 185-bp DNA fragments containing the Widom 601 spacing sequence, as described previously[54]. The 185-bp (final 6.1 μM) or 147-bp (final 5.6 μM) DNA fragment was mixed with a histone octamer at a molar ratio of ~1:1.1 and then dialyzed against 125 mL of 10 mM HEPES-Na pH 7.5, 2 M KCl, and 1 mM DTT for 1 h. Thereafter, 875 mL of 10 mM HEPES-Na pH 7.5 and 1 mM DTT was gradually added to facilitate nucleosome reconstitution. The nucleosome samples were further dialyzed against 10 mM HEPES-Na pH 7.5, 50 mM KCl, and 1 mM DTT. Finally, the centrifuged supernatant was concentrated and stored at 4 °C.

## Electrophoretic analysis of the Bre1-nucleosome complex

To monitor the formation of the Bre1-nucleosome complex, the nucleosome and trimeric Bre1 complex were mixed at different molar ratios and NaCl concentrations in 20 mM HEPES-Na at pH 7.5, 1 mM DTT, and 0.5 mM EDTA. The mixtures were incubated for an hour at 4 °C and then analyzed using electrophoresis on non-denaturing 4% acrylamide gel in 0.5% TBE buffer at 150 V for 40 minutes. The gels were stained with SYBR Gold (Thermo Fisher).

## Ubiquitination assay

For the ubiquitination assay using the trimeric Bre1A-Bre1B-Rad6A complex, 1.2 µM 147- or 185-bp nucleosome, 100 nM E1, 0.5–1.5 µM Bre1 complex, and 36 µM ubiquitin were mixed in 50 mM Tris-HCl at pH 8.0, 50 mM NaCl, 50 mM KCl, 3 mM ATP, 10 mM $MgCl_2$, and 1 mM DTT and incubated at 30 °C for 90 min. For the ubiquitination assay of the truncated RING[A]-RING[B] heterodimers, 1 µM 147-bp nucleosome, 100 nM E1, 10 µM RING[A]-RING[B] heterodimer or its mutants, and 36 µM ubiquitin were mixed in 50 mM Tris-HCl at pH 8.0, 125 mM NaCl, 3 mM ATP, 10 mM $MgCl_2$, 1 mM DTT, and incubated at 30 °C for 1, 2, or 4 hours. Reactions were stopped by the addition of 3× SDS loading buffer and analyzed by Western blotting or using Wes (ProteinSimple). For Western blotting, Ubiquityl-Histone H2B (Lys120) (D11) Rabbit mAb (No. 5546; Cell Signaling) was used as the primary antibody (1:1000 dilution), goat anti-rabbit IgG-HRP (sc-2004; Santa Cruz Biotechnology) as the secondary antibody (1:2000 dilution), and ECL Prime (Cytiva) as the chemiluminescent reagent. The images were recorded using LAS-3000 mini (Fujifilm). For Wes analysis, Ubiquityl-Histone H2B (Lys120) (D11) Rabbit mAb was used as the primary antibody (1:250 dilution), and Anti-Rabbit Detection Module (DM-001; ProteinSimple) was used without dilution following the manufacturer's protocol. As the normalization control, a single reaction mix using the wide type RING[A]-RING[B] heterodimer was created in a large volume and loaded each time. The data were quantified using Compass for SW v 6.2.0 (ProteinSimple).

## Microscale thermophoresis

The microscale thermophoresis assay was performed using a Monolith NT.115 instrument (NanoTemper Technologies). For the MST assay, the His6-tagged RING domain heterodimers (200 nM) labeled with RED-Tris-NTA and a series of twofold-diluted nucleosome or DNA (from 0.82 nM to 26.5 µM) were incubated in the binding buffer (10 mM HEPES-Na pH 7.5, 25 mM KCl, 10% glycerol, 0.05% Tween 20, and 1 mM DTT) for 30 min at room temperature and centrifuged at 15,000 × g for 10 min. Premium capillaries were filled with the samples to obtain measurements with an extinction power of 80-100% and medium MST power at 25 °C. Thermophoresis data were analyzed using MO. Affinity Analysis software ver. 2.3 (NanoTemper Technologies).

## Cryo-EM sample preparation and data acquisition

Th nucleosome (147 bp DNA) and Bre1 complex (Bre1A-Bre1B-Rad6A) were mixed in a 1:4 molar ratio. Final concentrations of 4.5 µM NCP(400 µg/mL dsDNA) and 18 µM BRE1 complex were incubated in buffer (20 mM HEPES at pH 7.5, 30 mM NaCl, 1 mM DTT, and 1 mM EDTA) at 4 °C for an hour. The sample was then diluted three times, and 3 µl of the diluted sample was applied onto Cu-Rh grids (quantifoil) and plunge-frozen in liquid ethane using Vitrobot Mark IV (Thermo Fisher Scientific). The images were obtained with a 300 kV Titan Krios G3i microscope (Thermo Fisher Scientific) equipped with a K3 direct electron detector (Gatan), installed at the University of Tokyo, Japan. The data sets were acquired with the SerialEM software[55], with a defocus range of −0.8 to −1.8 µm. The data acquisition statistics are shown in Table 1.

## Cryo-EM data processing

The cryo-EM data were processed using cryoSPARC v4.2.1[56] (Supplementary Fig. 2). Particles were automatically picked with a blob picker module using 500 motion-corrected, CTF-estimated micrographs, and then subjected to multiple rounds of two-dimensional (2D) classification. The resultant particles were used for the training by Topaz[57], which was used to pick particles from all micrographs. After several rounds of 2D classifications on the Topaz-picked particles, multiple rounds of ab initio reconstruction and heterogenous refinement with cryoSPARC were conducted to remove junk particles. The resultant particles were used for Non-uniform refinement followed by 3D classification without focus mask to be classified into 10 classes, of which two showed a clear density for the two RING domains of the Bre1 complex. The particle belonging to the two classes (83,434 particles) were combined and subjected to Non-uniform refinement, yielding a 2.82 Å map of the Bre1-nucleosome complex. The particles belonging to the other eight classes (341,993 particles) were combined, subjected to a 2D classification followed by Non-uniform refinement, yielding 2.51 Å map of the free nucleosome. The resolution was estimated based on the gold standard Fourier shell correlation curve (FSC) at a 0.143 criterion.

## 3DFlex analysis

The 3DFlex analysis was performed using CryoSPARC. For the 3DFlex training, the particles were downsampled to a box size of 128 pixels (from 320 pixels), with the Nyquist limit of 5.25 Å. A tetrahedral mesh was created using default parameters. The training was performed with two latent dimensions and the overall rigidity of 0.1. The 3DFlex map used for the model building was calculated with the trained model and the original, non-downsampled particles.

## Model building and refinement

A model structure of the RING[A]-RING[B] heterodimer (calculated using AlphaFold2) and the nucleosome structure (PDB 1KX5) were fit into the density using UCSF Chimera[58] and then manually modified with Coot[59]. Real-space refinement was performed using the phenix.real_space_refine module in Phenix against the 3DFlex map for the Bre1-nucleosome complex and against the Non-uniform map (without sharpening) for the free nucleosome, with the nonbonded_weight parameter set to 200 to reduce clashes.

## Figure preparation

Structural figures were prepared using PyMOL v2.5.0 (https://pymol.org/2/) and ChimeraX v1.4[60].

## Reporting summary

Further information on research design is available in the Nature Portfolio Reporting Summary linked to this article.

## Data availability

The cryo-EM density maps were deposited to the Electron Microscopy Data Bank under accession code EMD-34274 and EMD-34275. The atomic coordinates were deposited to the Protein Data Bank under accession code 8GUI (Model I), 8GUJ (Model II), and 8GUK (Free nucleosome). Source data are provided with this paper.

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

## Acknowledgements

We thank Dr. Yusuke Sato (Tottori University), who kindly provided the plasmids encoding human E1 and ubiquitin. This study was supported by a grant from the Japan Society for the Promotion of Science KAKENHI (grant No. 21H05161) to T.S. The study was partially supported by the Platform Project for Supporting Drug Discovery and Life Science Research (Basis for Supporting Innovative Drug Discovery and Life Science Research), funded by the Japan Agency for Medical Research and Development (grant No. JP20am0101115, support No. 1061).

## Author contributions

T.S. conceived the project. S.O., K.U., C.O., S.K., and T.S. prepared the samples. S.O., K.S., T.N., O.N., and T.S. collected cryo-EM images. S.O., K.S., K. H., and T.S. solved the cryo-EM structures, with T.N.'s advice. S.O., K.U., C.O., and T.S. performed the biochemical analyses. S.O., K.U., K.O., and T.S. wrote the manuscript.

## Competing interests

The authors declare no competing interests.
