## [Peer Review File · Nature Communications]

Structure of the human Bre1 complex bound to the nucleosomeREVIEWER COMMENTS

Reviewer #1 (Remarks to the Author):

The Bre1 protein complex in humans is a hetero-dimeric ubiquitin E3 ligase composed of Bre1A and Bre1B proteins. This complex is responsible for targeting the Rad6A E2 ubiquitin conjugating enzyme to mono-ubiquitinate histone H2BK120 in humans (corresponding to H2B-K123 in yeast). This manuscript described a high-resolution cryo-EM structure of human Bre1 complex bound to the nucleosome. The structure revealed that the RING domains of complex interact with the acidic patch and the DNA phosphates near SHL 6 of the nucleosome. The Rad6A protein, present in the biochemical preparation of the complex, was not resolved in the density map, due to its flexibility. Using the cryo-EM map, two atomic models were constructed where Bre1A and Bre1B are on opposite orientations. These models, combined with complementary mutagenesis studies, allowed the authors to propose a comprehensive model for H2BK120-specific ubiquitination by the Bre1 complex. While this study demonstrates high-quality research, it is important to note that its novelty and significance may have been overshadowed by a similar study published earlier this year by the Wolberger lab in bioRxiv (doi: <https://doi.org/10.1101/2023.03.27.534461>). The authors should cite the preprint in the manuscript and also highlight both commonalities and differences between the two studies. Additional critiques are listed below.

Major concerns:

1. The authors should provide a better micrograph in Extended Data Fig 2a. The current micrograph exhibits low contrast and significant ice contaminations, making it difficult to discern individual particles. Additionally, the selected 2D classes in Extended Data Fig 2b appear to only show free nucleosomes in different orientations. It is essential for the authors to present 2D classes that clearly shows the additional density of the bound Bre1 heterodimer on the nucleosome. Both the micrograph and 2D class averages provide crucial information for readers to assess the dataset's quality.
2. In Extended Data Fig 6, the author concluded that DNA regions near SHL6 undergoes a small shift (3 Å) upon Bre1 binding. The conclusion was based on comparison of the atomic coordinates of Bre1-bound nucleosome with that of the free nucleosome. I caution against such interpretation for two reasons. Firstly, the resolutions of the terminal DNAs on both cryo-EM maps are limited (~4-5 Å as indicated in the local resolution maps in Extended Data Fig 3c,d,f), which is common in cryo-EM structures of nucleosomes due to DNA breathing near the entry/exit sites. The limitation and the inherent flexibility of nucleosomal DNA should be considered, as they hinder the precise determination of terminal DNA locations in that region. Consequently, measuring small structural differences in DNA in that region becomes challenging and may introduce biases. Secondly, the small conformational differences in DNA shown in Extended Data Fig6 are negligible, particularly when considering the abovementioned considerations. If the authors believe the shift is significant, they should provide more information about the comparison to support their claim. This may include showing the atomic coordinates fitted into their respective density maps in that region, presenting local B-factors of the map, and describing how errors in model-building and measurements were considered.

Minor concerns:

1. On page 3, paragraph 3 Line 6, "H2B120" should be "H2BK120".
2. The authors utilized color-coded histones in several figures (Fig 1, 3&5, Extended Data Fig 6 &8) to represent the nucleosome density map. However, they did not provide labels for each color, which could enhance the manuscript's readability. The authors should specify which histone each color represents in the figures.
3. Although movies 2&3 show the conformational changes along the two axes, the results from 3DFlex in Fig 2 should be presented as a series of convected densities, with a specific region in focus, rather than a comparison of two atomic coordinates, as shown in the current version.
4. The Discussion section in the current version needs to be expanded. For instance, the authors can discuss the study from the Wolberger lab in this section. Additionally, the content under the sub-title "Comparison with H2A ubiquitin ligases" may be more suitable for the Discussion. I also suggest moving Fig 6 to the Extended Data section.
5. The authors should provide validation reports for all their depositions in EMDB and PDB, for manuscript review purposes.
6. Several sections (summary, introduction, and discussion) of the current manuscript contain general statements about Bre1 proteins in cancers. While there are benefits of drawing connections to cancer, it is unclear what specific cancer-related questions the authors aim to explore. Furthermore, the current study does not directly address any questions regarding the Bre1 complex or H2BK120ub in cancer biology. The authors should either explicitly state the specific questions directly linked to the current study or modify the text accordingly.

Reviewer #2 (Remarks to the Author):

Onishi, et al. determined a cryo-EM structure of Bre1 heterodimer bound to nucleosome in order to gain insights into a mechanism of H2B ubiquitination. The structure reveals the position of the two RING domains on the nucleosome structure, showing binding to the acidic patch and nucleosomal DNA. In addition, the authors propose the mechanism of E2 (Rad6A) recruitment by Bre1 using modeling and mutations coupled to activity assays.

The paper is interesting. The authors combine medium-resolution cryo-EM data, modeling, and biochemistry experiments to reach their conclusions. In general, the statements are well supported by data. This paper could be significantly improved if the authors complemented their data with some crosslinking mass spectrometry and enzyme kinetics.

My suggestion is to lower the paper's tone to more mechanistic and address the comments below.

Main concerns:

- The relatively low resolution of the RING domain regions makes it difficult to model sidechains at interacting regions and discriminate between two possible binding orientations. Crosslinking the samples before EM should solve some resolution and missing subunit issues.
- Interaction with DNA and the acidic patch could be addressed by XLMS, which would validate the interactions and resolve the orientation problem in the binding mechanism.
- Authors claim that their data reveal the structural basis of ubiquitination. However, since Ubiquitin and Rad6A are not visible in their structure and must be modeled, the paper shows more “insights into mechanism” rather than “structural basis.”
- Mutation and modification of histones at the DNA entry are weak evidence for regulating Bre1 activity by DNA flexibility since flexibility is also observed without mutations. Plenty of structures of nucleosome complexes and nucleosomes alone at different salts show variable organization of ~10—13bp of exiting DNA.
- Can authors discuss more the activity assay where double mutants of RINGB and RINGA show an increase in activity over the WT? Shouldn't this mutant behave like WT if the model in 5 is correct?
- Following the previous point, the effect can be due to increased affinity, better catalytic efficiency, or better binding to the other side of the nucleosome. A kinetic analysis will be needed to clarify the mechanism. Binding affinity experiments for those mutants on nucleosomes should also clarify this issue to some extent.

Minor:

- in the section “bre1A as a catalytic subunit,” when threonine mutants are named T948GA-T952GA, the T is missing.
- In figure 5A, colors are slightly tricky to follow.
- “Supplementary Movie 1 Electron density map and overall structure of Model I” should be cryo-EM density instead
- Figure 2, please specify the movie numbers.
- Extended Fig. 4. Please label arginines in panel I.

We thank all the reviewers for their very helpful comments. We have revised the manuscript according to the reviewers' suggestions. We also changed the explanation of the contact region between Bre1 and nucleosomal DNA (from SHL 6.0 to SHL 6.0-6.5) to be more precise.

Our point-to-point responses are shown below in blue.

Reviewer #1 (Remarks to the Author):

The Bre1 protein complex in humans is a hetero-dimeric ubiquitin E3 ligase composed of Bre1A and Bre1B proteins. This complex is responsible for targeting the Rad6A E2 ubiquitin conjugating enzyme to mono-ubiquitinate histone H2BK120 in humans (corresponding to H2B-K123 in yeast). This manuscript described a high-resolution cryo-EM structure of human Bre1 complex bound to the nucleosome. The structure revealed that the RING domains of complex interact with the acidic patch and the DNA phosphates near SHL 6 of the nucleosome. The Rad6A protein, present in the biochemical preparation of the complex, was not resolved in the density map, due to its flexibility. Using the cryo-EM map, two atomic models were constructed where Bre1A and Bre1B are on opposite orientations. These models, combined with complementary mutagenesis studies, allowed the authors to propose a comprehensive model for H2BK120-specific ubiquitination by the Bre1 complex. While this study demonstrates high-quality research, it is important to note that its novelty and significance may have been overshadowed by a similar study published earlier this year by the Wolberger lab in bioRxiv (doi: <https://doi.org/10.1101/2023.03.27.534461>). The authors should cite the preprint in the manuscript and also highlight both commonalities and differences between the two studies. Additional critiques are listed below.

Major concerns:

1. The authors should provide a better micrograph in Extended Data Fig 2a. The current micrograph exhibits low contrast and significant ice contaminations, making it difficult to discern individual particles. Additionally, the selected 2D classes in Extended Data Fig 2b appear to only show free nucleosomes in different orientations. It is essential for the authors to present 2D classes that clearly shows the additional density of the bound Bre1 heterodimer on the nucleosome. Both the micrograph and 2D class averages provide crucial information for readers to assess the dataset's quality.

We have replaced Extended Data Fig. 2a with a new micrograph with 5 Å lowpass-filter to enhance contrast. Almost all of the images in our dataset are contaminated with ice. Thus, although the new micrograph also contains contaminations, it fairly represents the data quality.

Extended Data Fig 2b shows the 2D classification analysis on the mixture Bre1-free and Bre1-bound nucleosomes, which were separated later by further processing. We retain this figure in the revised manuscript to faithfully explain how we analyzed the dataset. We also think it helpful for readers to show an example where, even the initial 2D averages contain no clearly recognizable complex

images, further processing sometimes results in the successful separation of the small amount of the complex. To show the quality of the separated particles belonging to the Bre1-bound nucleosomes, we performed a retrospective 2D classification on them, where density of the bound Bre1 molecule is visible (Extended Data Fig 2c).

2. In Extended Data Fig 6, the author concluded that DNA regions near SHL6 undergoes a small shift (3 Å) upon Bre1 binding. The conclusion was based on comparison of the atomic coordinates of Bre1-bound nucleosome with that of the free nucleosome. I caution against such interpretation for two reasons. Firstly, the resolutions of the terminal DNAs on both cryo-EM maps are limited (~4-5 Å as indicated in the local resolution maps in Extended Data Fig 3c,d,f), which is common in cryo-EM structures of nucleosomes due to DNA breathing near the entry/exit sites. The limitation and the inherent flexibility of nucleosomal DNA should be considered, as they hinder the precise determination of terminal DNA locations in that region. Consequently, measuring small structural differences in DNA in that region becomes challenging and may introduce biases. Secondly, the small conformational differences in DNA shown in Extended Data Fig6 are negligible, particularly when considering the abovementioned considerations. If the authors believe the shift is significant, they should provide more information about the comparison to support their claim. This may include showing the atomic coordinates fitted into their respective density maps in that region, presenting local B-factors of the map, and describing how errors in model-building and measurements were considered.

We agree with the reviewer in that the current data is not sufficient to claim that the structural shift observed between the Bre1-free and Bre1-bound nucleosomes is functionally important. We have added a sentence to make it clear that we are not confident about the functional importance of this structural shift.

“Since the local resolution of this DNA region is relatively low, as often observed in the nucleosome structures, the functional significance of this small shift is unclear.”

Minor concerns:

1. On page 3, paragraph 3 Line 6, “H2B120” should be “H2BK120”.

We have corrected this typo.

2. The authors utilized color-coded histones in several figures (Fig 1, 3&5, Extended Data Fig 6 &8) to represent the nucleosome density map. However, they did not provide labels for each color, which could enhance the manuscript's readability. The authors should specify which histone each color represents in the figures.

We have added labels in Fig. 1, 3, and 5, and Extended Data Fig 6, 8, 9, and 12 to show colors used in each histone.

3. Although movies 2&3 show the conformational changes along the two axes, the results from 3DFlex in Fig 2 should be presented as a series of convected densities, with a specific region in focus, rather than a comparison of two atomic coordinates, as shown in the current version.

Figures 2b-e show the superposition of atomic coordinates. The purpose of these figures is to illustrate the structural heterogeneity of the particles as revealed by the 3D-Flex analysis. We are concerned that the superposition of two densities is too complicated and difficult to interpret. We assume that the reviewer requires the densities for quality evaluation. For this purpose, we have created separate density figures as extended data figures and kept the original in the main figure.

4. The Discussion section in the current version needs to be expanded. For instance, the authors can discuss the study from the Wolberger lab in this section. Additionally, the content under the sub-title "Comparison with H2A ubiquitin ligases" may be more suitable for the Discussion. I also suggest moving Fig 6 to the Extended Data section.

Following this comment, we have discussed the study on the yeast Bre1-nucleosome structure (Zhao et al. bioRxiv 2023, by the Wolberger lab), and also the recent study on the yeast and human Bre1-nucleosome structures (Deng et al. Mol Cell 2023, by the Liu lab). The three studies revealed essentially the same interactions between the two Bre1 subunits and the nucleosome; one subunit binds stably to the acidic patch via arginine anchors, and the other subunit binds flexibly to the nucleosomal DNA. A notable difference is how they and we prepared the samples. Our study used the native full-length proteins for the structure determination, hopefully with minimal structural artifact, while Deng et al. used very complex chemical tricks to stabilize the complex, leading to the visualization of the tethered Rad6 density. Zhao et al. also used a chimeric protein consisting of the two tandem RING domains of Bre1 linked to Rad6. Thus, the three studies complement each other to confirm the specific nucleosome recognition by the Bre1 proteins that is conserved between human and yeast. We also note that Deng et al. used an asymmetrically modified human Bre1-Rad6A complex, with the Bre1A/RNF20 subunit fused to Rad6A via a 30-mer linker and modeled the Bre1 heterodimer in one orientation (Bre1A/RNF20 bound to the acidic patch), apparently without decisive experimental support. Our experimental results suggest that the heterodimer binds the nucleosome in two orientations and that the ubiquitination reaction occurs when Bre1A is bound to the acidic patch. Thus, our study provides evidence that the Bre1-Rad6A-nucleosome model by Deng et al. is likely to be catalytically competent and collectively reveals the detailed insight into the H2BK120-specific ubiquitination mechanism.

We have also moved the chapter on the comparison with H2A ubiquitin ligases to the Discussion and Fig. 6 to the Extended Data section (Extended Data Fig. 12).

5. The authors should provide validation reports for all their depositions in EMDB and PDB, for manuscript review purposes.

We have provided validation reports of the three deposited structures.

6. Several sections (summary, introduction, and discussion) of the current manuscript contain general statements about Bre1 proteins in cancers. While there are benefits of drawing connections to cancer, it is unclear what specific cancer-related questions the authors aim to explore. Furthermore, the current study does not directly address any questions regarding the Bre1 complex or H2BK120ub in cancer biology. The authors should either explicitly state the specific questions directly linked to the current study or modify the text accordingly.

We agree with the reviewer in that the original manuscript lacked the specific research question regarding the relationship between Bre1 and cancer. We have revised the Abstract (Summary), Introduction, and Discussion sections to soften our statement about the association of our study with cancer.

Reviewer #2 (Remarks to the Author):

Onishi, et al. determined a cryo-EM structure of Bre1 heterodimer bound to nucleosome in order to gain insights into a mechanism of H2B ubiquitination. The structure reveals the position of the two RING domains on the nucleosome structure, showing binding to the acidic patch and nucleosomal DNA. In addition, the authors propose the mechanism of E2 (Rad6A) recruitment by Bre1 using modeling and mutations coupled to activity assays.

The paper is interesting. The authors combine medium-resolution cryo-EM data, modeling, and biochemistry experiments to reach their conclusions. In general, the statements are well supported by data. This paper could be significantly improved if the authors complemented their data with some crosslinking mass spectrometry and enzyme kinetics.

My suggestion is to lower the paper's tone to more mechanistic and address the comments below.

Main concerns:

- The relatively low resolution of the RING domain regions makes it difficult to model sidechains at interacting regions and discriminate between two possible binding orientations. Crosslinking the samples before EM should solve some resolution and missing subunit issues.

We had initially tried to crosslink the EM sample to stabilize the complex during structure determination. However, a native gel electrophoresis showed that when a crosslinking reagent was added to the mixture, the sample became highly heterogeneous (bands smeared in the gel) compared to the non-crosslinked sample (Extended Data Fig. 1b). Therefore, we decided not to use crosslinking in EM sample preparation.

- Interaction with DNA and the acidic patch could be addressed by XLMS, which would validate the interactions and resolve the orientation problem in the binding mechanism.

We had initially tried crosslinking mass spectrometry to determine which of the two orientations of the Bre1 complex is predominant. However, we were unable to detect any histone-Bre1 cross-link product by mass spectrometry, possibly because of the weak interactions between the Bre1 complex and the nucleosome.

- Authors claim that their data reveal the structural basis of ubiquitination. However, since Ubiquitin and Rad6A are not visible in their structure and must be modeled, the paper shows more “insights into mechanism” rather than “structural basis.”

Following this comment, we have revised the manuscript to use the phrase "structural insight into...".

- Mutation and modification of histones at the DNA entry are weak evidence for regulating Bre1 activity by DNA flexibility since flexibility is also observed without mutations. Plenty of structures of nucleosome complexes and nucleosomes alone at different salts show variable organization of ~10–13bp of exiting DNA.

In the original manuscript, we had failed to cite an experiment showing that a linker histone inhibits H2BK120 ubiquitination (Wojcik et al. 2018). Linker histones are known to stabilize the conformation of nucleosomal DNA, so this result provides additional support for the idea that DNA flexibility regulates H2BK120 ubiquitination. We have cited this experiment in the revised manuscript and rewrote the chapter "Bre1 activity may be regulated by the flexibility of nucleosomal DNA" accordingly. As stated in the last sentence of the chapter, we acknowledge that this hypothesis should be tested by further studies, but we still think that it is worth proposing.

- Can authors discuss more the activity assay where double mutants of RINGB and RINGA show an increase in activity over the WT? Shouldn't this mutant behave like WT if the model in 5 is correct?

We think that the RINGB TT mutant has somehow gained a higher activity than the wild type, and in the double mutant the effect of the RING^A mutations is masked by the high activity of the RING^B TT mutant. We have discussed this in the text.

- Following the previous point, the effect can be due to increased affinity, better catalytic efficiency, or better binding to the other side of the nucleosome. A kinetic analysis will be needed to clarify the mechanism. Binding affinity experiments for those mutants on nucleosomes should also clarify this issue to some extent.

We performed microscale thermophoresis analysis to measure the binding affinities of the wild-type and mutant Bre1 constructs. The results showed that these mutants did not exhibit increased nucleosome affinity, suggesting that the gain in the activity of the TT mutants is due to their ability to recruit Rad6A. Our ubiquitination assay uses antibody-based detection coupled to capillary electrophoresis, and it is very difficult to determine kinetic parameters with this method due to its low throughput. Therefore, we decided not to perform a kinetic analysis and believe that our biochemical experiments support the proposal that Bre1A is the likely catalytic subunit.

Minor:

- in the section "bre1A as a catalytic subunit," when threonine mutants are named T948GA-T952GA, the T is missing.

We have corrected this error. We also refer to T948G^A-T952A^A as GA and G974T^B-A978T^B as TT in the revised manuscript for simplicity.

- In figure 5A, colors are slightly tricky to follow.

We have revised this figure.

- "Supplementary Movie 1 Electron density map and overall structure of Model I" should be cryo-EM density instead

We have corrected this error.

- Figure 2, please specify the movie numbers.

We have added the Supplementary Movie numbers.

- Extended Fig. 4. Please label arginines in panel I.

We have added labels in Extended Fig. 4i.

REVIEWERS' COMMENTS

Reviewer #1 (Remarks to the Author):

The authors have adequately addressed the concerns raised in the initial review. Upon examining the clashscores outlined in the PDB validation reports for all three models, I noticed that these scores are unexpectedly high, especially considering the high resolution of the maps. Although these scores do not undermine the conclusions drawn in the manuscript, optimizing the quality of the models deposited in the PDB would significantly benefit both the authors and the broader scientific community. I recommend that the authors consider refining their models to further enhance their quality before finalizing their deposition.

Reviewer #2 (Remarks to the Author):

I am happy with the revision and recommend publication

We thank all the reviewers for their favorable comments. Our point-to-point response is shown below in blue.

Reviewer #1 (Remarks to the Author):

The authors have adequately addressed the concerns raised in the initial review. Upon examining the clashscores outlined in the PDB validation reports for all three models, I noticed that these scores are unexpectedly high, especially considering the high resolution of the maps. Although these scores do not undermine the conclusions drawn in the manuscript, optimizing the quality of the models deposited in the PDB would significantly benefit both the authors and the broader scientific community. I recommend that the authors consider refining their models to further enhance their quality before finalizing their deposition.

Following this comment, we performed another round of refinement using phenix.real_space_refine with the nonbonded_weight parameter set to 200 (instead of the default value of 100) for the three structures (description added in the Methods section). In the re-refined structures, the clash scores are reduced from 9.43 to 6.55 (model I), from 9.59 to 7.41 (model II), and from 10.18 to 5.13 (free nucleosome), with no major structural changes. We upload the validation reports for the three re-refined structures. We thank the reviewer for the helpful comment.